# The distribution of different intensity demanding scenarios in elite rink hockey players using an electronic performance tracking system

Daniel Fernández[1]*, Joan A. Cadefau[2], Noemí Serra[3], Gerard Carmona[3]*

**1** Sport Science Department, Futbol Club Barcelona, Barcelona, Spain, **2** Institut Nacional d'Educació Física de Catalunya (INEFC), Universitat de Barcelona (UB), Barcelona, Spain, **3** Departament de Ciències de la Salut Tecnocampus, Universitat Pompeu Fabra, Mataró, Spain

* daniel.fernandez@fcbarcelona.cat (DF); gcarmona@tecnocampus.cat (GC)

## Abstract

Despite the traditional use of average values for determining physical demands, the intermittent and fluctuating nature of team sports may lead to underestimation of the most demanding scenarios. All the most demanding scenario-related investigations to date only report one maximal scenario per game, the greatest. However, the latest research on this subject has shown additional scenarios of equal or similar magnitude that most researchers have not considered. This repetition concept started a new way of describing competition and training loads; then the study aims were: first, to quantify and assess differences between playing positions in terms of the most demanding scenarios in official matches; and second, to quantify and assess the differences between playing positions in the repetition of different intensity scenarios relative to the most demanding individual scenario. We monitored nine professional rink hockey players (7 exterior and 2 interior players) in 18 competitive matches using an electronic performance tracking system. The interior players are closest to the opponent's goal, while the exterior players are farthest from it. Peak physical demands variables included total distance (m), distance covered at >18 km·h⁻¹ (m), the number of accelerations ($\geq$2 m·s⁻², count) and decelerations ($\leq$-2 m·s⁻², count) in 30 s. An average from the top three individual most demanding scenarios was used to define a reference value to quantify the distribution scenario repetition during matches. The results showed that peak demands in rink hockey are position-dependent, with more distance covered by exterior players and more accelerations performed by interior players. In addition, rink hockey matches include multiple scenario exposures that are close to the peak physical demands of a match. Using the results of this study, coaches can prepare tailored training plans for each position, focusing on distances covered or accelerations for exterior players.

## Introduction

Rink hockey is a dynamic and complex indoor team sport in which players use roller skates to perform intermittent and multidirectional actions, such as accelerations, decelerations, sprints,

**Data Availability Statement:** All relevant data are within the paper and its Supporting information files.

**Funding:** This research was funded by TecnoCampus (Universitat Pompeu Fabra), grant number F027/2021. DF recieved that grant. The funder website is: https://www.tecnocampus.cat. The funders had no role in study design, data collection and analysis, decision to publish, or preparation of the manuscript.

**Competing interests:** The authors have declared that no competing interests exist.

and impacts. All these actions are performed in a sport-specific framework of driving, shooting, passing, blocking, and defending. Roller skating makes rink hockey a dynamic and fast sport that has relative average demands far greater than other indoor sports [1, 2]. Actually, there has been a trend in growth of the use of inertial devices combined with ultra-wide band (UWB) positioning technology to track distance, accelerations, and decelerations during matches and training sessions. Rink hockey has not escaped this, and information about physical and position-related demands can guide sports scientists in rink hockey in the improvement of training programmes and the use of accurate training loads that minimise the risk of injury [3]. Research comparing different team sports [4] showed that rink hockey along with soccer achieves the greatest values in total distance and distance above 18 km/h; and along with basketball achieves the greatest values in the number of accelerations and decelerations. Finally, other research on external loads in official rink hockey competitions has shown that no differences exist between playing positions (exterior and interior players) in the average physical demands imposed by matches [1, 2]. After that, coaches could use that information to plan and prepare the training microcycle, and the individual training sessions for players; as rink hockey is a high intensity sport without any differences between positions in terms of average demands.

However, the approach used to assess average physical demand underestimates peak intensities during indoor or outdoor matches due to the intermittent nature of team sports [5]. Some peak moments may be hidden by the average values of all game signals. For this reason, it is necessary to use other monitoring methods to determine the most demanding scenarios (MDS), a period of peak demands experienced by the players during a match, and to implement proper training intensities during sport-specific drills [6].

There has been a growth in research using rolling or moving averages [7] to examine peak competition demands in team sports based on identifying the MDS, also referred to as the most demanding passages in the literature [8, 9]. This new approach provides important results that may complement the previous research that mostly focussed on average physical demands. Monitoring the MDS using moving averages helps scientists and coaches understand the parameters that are more important in the analysis of team sports loads because this measure takes into account the naturally intermittent nature of team sports, where, due to the variability of the matches, it is expected that some moments will be more intense than others [10].

Research into peak demands during different epochs (frequently 30 and 60 s) has been conducted using rolling average methods in soccer [11], rugby [12], basketball [6, 13], futsal [14], and rink hockey [8]. To date, all the cited investigations only report one maximal scenario per game, with the latest MDS research [9, 15] showing additional scenarios of equal or similar magnitude that most researchers and sport scientists did not consider in a given match. The approach of only describing one MDS and not all the different intensity scenarios could underestimate the overall physical demands that occur in a match (e.g., it could be that a player must cope with more than 4 or 5, 30 or 60 s scenarios, near to the MDS). This concept was first described in the literature as the repetition of high (80%–90% of the MDS), or very high (>90% of the MDS) demanding scenarios [9]. Finally, the repetition of scenarios to study the distribution of physical activities in team sports is rare in sports science, and to the authors' knowledge, studies have only reported the distribution of maximal and submaximal demands relative to the peak intensities in basketball matches [16], futsal matches [9, 15], and rugby and Australian rules football matches [17, 18]. To date, no study has described the distribution of different scenarios of relative intensity in competitive rink hockey and this study aims to do it.

In this study, we have aimed to quantify and assess differences between playing positions in terms of the MDS in official matches. In addition, we have aimed to quantify and assess the

differences between playing positions in the repetition of different intensity scenarios relative to the individually most demanding scenario. Based on the previous literature in rink hockey, the hypothesis of this study was that interior players experience the same MDS and repetition of different intensity scenarios as the exterior players.

## Materials and methods

### Design

We conducted an observational retrospective study in a cohort with a time of follow-up of 2 years during the 2019–2020 and 2020–2021 competitive seasons. In 2019–2020, we were forced to stop in March 2020 due to COVID-19. External load was measured by a UWB electronic performance tracking system (WIMU PROTM, Realtrack Systems, Almeria, Spain) during 18 official matches of a professional team in the highest Spanish tier (Ok Liga), and that also participated in the WSE Rink Hockey EuroLeague.

### Subjects

Nine professional elite outfield players (age 29.0 ± 5.22 years, weight 78.8 ± 6.20 kg, height 180.6 ± 3.91 cm; all measurements in mean ± standard deviation) participated voluntarily. Goalkeepers did not participate in the study due the goalkeeper's own characteristics and the great technical difference compared to court players. Data from each player and match had to meet two inclusion criteria: (1) players finished the match without injury; and (2) players participated in the entire pre-match training session to ensure that we only included players who had prepared for the match the day previously. The nine players were comprised of seven exteriors and two interiors, as determined by three qualified rink hockey coaches (all of whom had national level certificate in rink hockey coaching from the Spanish Skating Federation, as well as at least three years' experience coaching teams in national leagues). In the possession phase, exterior players are those who play far from the opponent's goal with a lot of mobility, while interior players are those who play closest to the goal with less mobility and less space. To compare average per game scenarios between positions we checked that all analysed players did not have significant differences in the minutes played per game. The data which were analysed arose from routine daily monitoring of activities during the season. The ethics committee for clinical research of the Catalan sports council approved the experimental procedures used in this study (29/CEICGC/2020), and all the players provided informed consent before participating.

### Procedure

All matches were completed on the same official sport-specific indoor court in similar environmental conditions (court material, temperature, light, facilities. . .) and during in-season weeks. All players completed a standard 45-minute warm-up comprising dynamic mobility and individual sport-specific skills, such as passing, shooting, dribbling, and playing a small-sided game. Although the players were continuously monitored throughout the warm-ups and total match time, we only analysed data from when players were competing on court. We removed all resting data that was collected after substitutions and between periods.

Each player wore the same micro-technology positioning system device (81 × 45 × 15 mm, 70 g) throughout to reduce potential inter-unit variability [19]. The WIMU PRO™ units are equipped with four 3D accelerometers (full-scale output ranges are ± 16 g, ± 16 g, ± 32 g, ± 400 g, at 100 Hz sample frequency), three gyroscopes (8000º/s full-scale output range, at 100 Hz sample frequency), a 3D magnetometer (100 Hz sample frequency), a global positioning

system (10 Hz sample frequency), and a local positioning system with UWB technology (18 Hz sample frequency). For better signal emission and reception, six UWB antennas were installed on the rink hockey court to form a hexagon (with a sampling frequency of 18 Hz for the positioning data). Recently, the WIMU PRO™ system presented a high intra-class correlation coefficient value for the x-coordinate (0.65), a very high value for the y-coordinate (0.85), and a good percentage technical measurement error of 2.18% [20].

We included four physical demand parameters to describe the MDS and distribution of match activities. These included the following: distance travelled in meters (DT; m); high-speed skating; distance covered >18 km/h, in meters (HSS >18 km·h−1; m); number of high-intensity accelerations (ACC; >2 m·s−2; count); and number of high-intensity decelerations (DEC; >2 m·s−2; count). All these variables were chosen for two reasons: first, they were the ones most commonly used in other descriptive studies of competition loads in indoor sport research, and the thresholds used were the ones most commonly used in other external load research studies on rink hockey [1, 2, 4, 8], which allowed us to compare our results with previous literature, and second, it is a starting point in this kind of literature for further research on both variables and thresholds. After each match, data from the local positioning system were downloaded and extracted using the manufacturer's software (SPRO™, Realtrack Systems SL, version 977). The software provided instantaneous raw data as a rolling average for each variable and player over 30 s epochs. This specific epoch was chosen because it has already been used in previous research [5, 15], and could therefore be used to compare common average physical demands during competition.

We extracted the MDS data from the rolling average data for each player, variable, and match; and computed the distribution of the relative physical activities using a two-step process for each player and variable [9, 15]. At first, we determined an individual reference value (100% MDS reference value) from the mean of the top three observations per player using the 18 matches sample. That top three observations mean (instead of taking the maximal MDS recorded) had the main goal of avoiding any extreme value that was real in competition (because it was recorded) although it was too unrealistic to happen almost every match, whereas to smooth possible outliers. This is the reason why in the results section, some 30 s scenarios are above 100% of the individual MDS. These individual reference values are presented in Table 1 and compared with the mean MDS of all matches. In the second step, we re-analysed all matches searching all the 30 s scenarios in the rolling average data, transforming the absolute values to relative ones using the maximal reference values likewise in internal load studies the heart rate is expressed through relative values related to the individual's maximal heart rate. Thereafter, we established 10% buckets, counting the number of scenarios in each bucket. The final output was the total number of scenarios per each bucket, player, and variable. All the previous procedures are summarized in Fig 1.

## Statistical analysis

All statistical analyses were conducted with RStudio version 1.3.1093 (RStudio, Inc.). MDS descriptive data and the number of scenarios per bucket are reported as means ± standard deviations. Sample size analysis was not conducted due to the use of the bootstrap method approach and its assumptions about sample size (bootstrap is a suitable method when the sample size is small) [21]. We conducted an exploratory analysis using boxplots to identify possible outliers of the data. To assess the differences between positions, we used a bootstrap confidence interval approach for MDS and the number of scenarios per bucket [21, 22]. A re-sampling model, with 2,000 bootstrap samples and a 95% bias-corrected and accelerated method, was used to calculate the confidence intervals of $t$-test values for each variable. This established

**Table 1. Individual reference of the most demanding scenario values for each player extracted from all the games analysed, with the overall means, by position, in 30 s time epochs.**

| Position | Player | DT (m) | | HSS (m) | | ACC (count) | | DEC (count) | |
|---|---|---|---|---|---|---|---|---|---|
| | | Top 3 | all | Top 3 | all | Top 3 | all | Top 3 | all |
| Exterior | P1 | 126 | 119 | 62.7 | 49.5 | 6.67 | 5.83 | 7.00 | 5.83 |
| Exterior | P2 | 145 | 134 | 88.2 | 70.4 | 8.00 | 6.72 | 8.00 | 6.67 |
| Exterior | P3 | 142 | 132 | 81.2 | 65.1 | 6.67 | 6.00 | 6.00 | 5.50 |
| Exterior | P4 | 137 | 126 | 73.0 | 57.5 | 8.00 | 6.56 | 7.33 | 6.00 |
| Exterior | P5 | 147 | 132 | 93.7 | 67.7 | 6.33 | 5.44 | 6.00 | 5.00 |
| Exterior | P6 | 122 | 122 | 51.2 | 51.2 | 6.00 | 6.00 | 6.67 | 6.67 |
| Exterior | P7 | 137 | 124 | 77.3 | 55.0 | 6.67 | 5.89 | 6.33 | 5.67 |
| | mean | 136 | 127 | 75.3 | 59.5 | 6.90 | 6.06 | 6.76 | 5.90 |
| | SD | 9.46 | 5.79 | 14.6 | 5.25 | 0.79 | 0.44 | 0.74 | 0.61 |
| Interior | P8 | 136 | 124 | 71.2 | 54.2 | 7.33 | 6.61 | 7.33 | 6.11 |
| Interior | P9 | 132 | 126 | 76.0 | 60.5 | 7.67 | 6.50 | 7.33 | 6.06 |
| | mean | 134 | 125 | 73.6 | 57.4 | 7.50 | 6.56 | 7.33 | 6.09 |
| | SD | 2.69 | 1.53 | 3.42 | 4.44 | 0.24 | 0.08 | 0.00 | 0.03 |

Note. Data shows mean of the top 3 MDS results for each player with the overall means and SD by position. Abbreviations: ACC, high-intensity accelerations; DEC, high-intensity decelerations; DT, distance travelled; HSS, high-speed skating; MDS, most demanding scenarios; SD, standard deviations.

the null hypothesis, that there were no differences between positions, and the mean difference between positions was computed and presented as standardised differences (Cohen's *d*). Thresholds for standardised differences were <0.20) for trivial, 0.20–0.59 for small; 0.60–1.19 for moderate, 1.20–1.99 for large, and >2.0 for very large [23]. All reported p-values reflect the likelihoods of observing the absolute effect sizes assuming a true null hypothesis of zero

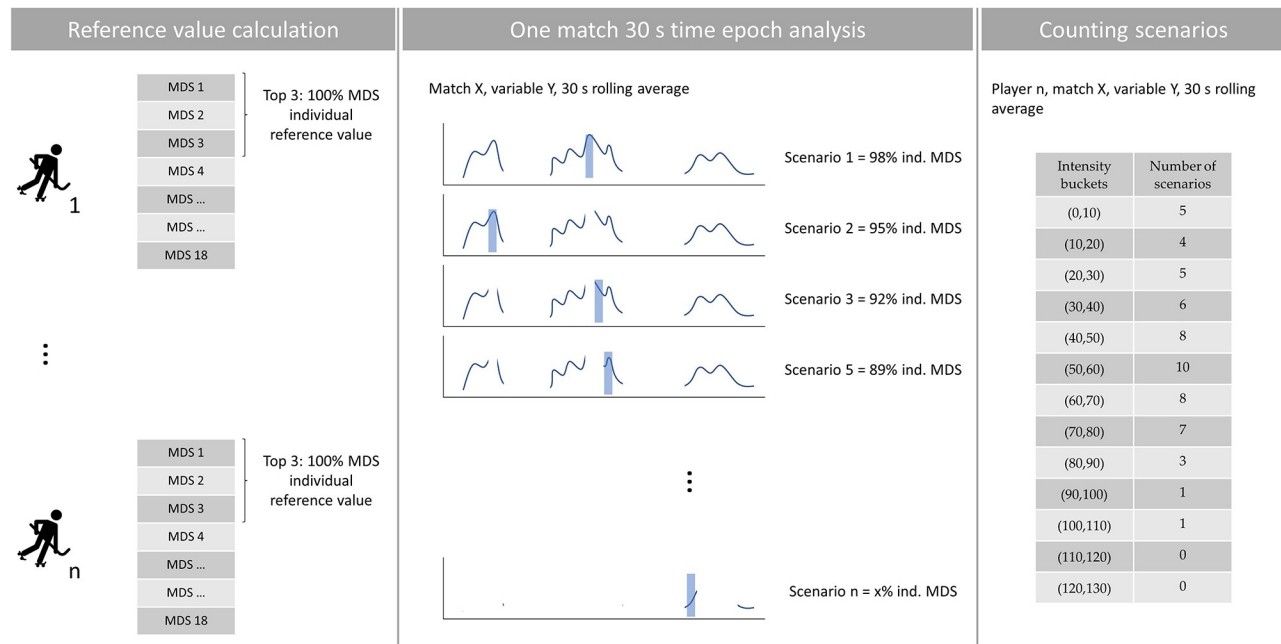

**Fig 1. Brief schematic representation of methodological procedures used to calculate: The individual reference values (left panel), all the scenarios in one match (central panel), and how the scenarios are counted (right panel).** Abbreviations—MDS: most demanding scenario.

difference [24]. Differences are presented with Cohen's *d* for the effect size and their corresponding 95% confidence intervals.

## Results

Table 2 summarises the descriptive results for the MDS, and Fig 2 shows the hypothesis test results for the MDS. Interiors had greater ACC with significant but small effect sizes.

Fig 3 and Table 3 show the descriptive results for the number of scenarios per bucket and variable per match. The DT variable had a greater number of repetitions in the central area of the buckets (between 60%–70% and 70%–80%); the HSS variable had a repetition peak in the first buckets; and the ACC and DEC variables had the greatest number of the scenarios between the 10% and 40% buckets.

Standardised differences (Cohen's d) and 95% confidence intervals between playing positions are presented in the Fig 4 for all variables and buckets. Exterior players did significantly more scenario repetitions in some buckets for the HSS variable, but with a small effect size. By contrast, both interiors and exteriors showed significant moderate differences in repetition scenarios for the ACC and DEC variables in different buckets and in different and unclear directions.

## Discussion

The present study aimed to quantify and assess differences between playing positions in terms of the MDS in official matches. In addition, it aimed to quantify and assess the differences between playing positions in the repetition of different intensity scenarios relative to the individually MDS. We found that the MDS demands during elite competition, although small to trivial, differed by player position in the ACC variable. In addition, rink hockey players were found to be exposed to different intensity 30 s scenarios that repeated more than once depending on their intensity and the metric analysed; exterior players did significantly more scenario repetitions in some buckets for the HSS variable, but with a small effect size.

### Quantification of the MDS and the differences between positions

A major finding of this study was the quantification of peak physical demand, understood as the MDS in competition. The activity profile described in the current investigation and elsewhere [5, 6] seems much more demanding than reported in a previous rink hockey study [2] that used average measures instead of rolling average epochs. The latter study found that players averaged 129–132 m·min$^{-1}$ for DT, 24.8–26.8 m·min$^{-1}$ for HSS, 4.8 count·min$^{-1}$ for ACC, and 4.5 count·min$^{-1}$ for DEC. Our investigation indicates that rink hockey had higher MDS values in 30 s time epochs compared to all these averages. In accordance with Vázquez-Guerrero et al. [6], our results improve coaches and sport scientists' understanding of the demands

**Table 2. Most demanding scenarios descriptive results by variable and position.**

| Epoch | Variable | Exterior | Interior |
|---|---|---|---|
| 30 s | DT (m) | 127 ± 9.28 | 125 ± 6.23 |
| | HSS (m) | 60.5 ± 14.3 | 57.2 ± 10.4 |
| | ACC (count) | 6.07 ± 0.87 | 6.56 ± 0.75 |
| | DEC (count) | 5.81 ± 0.89 | 6.09 ± 0.90 |

Note. Data show means ± standard deviations. Abbreviations: ACC, high-intensity accelerations; DEC, high-intensity decelerations; DT, distance travelled; HSS, high-speed skating; MDS, most demanding scenarios.

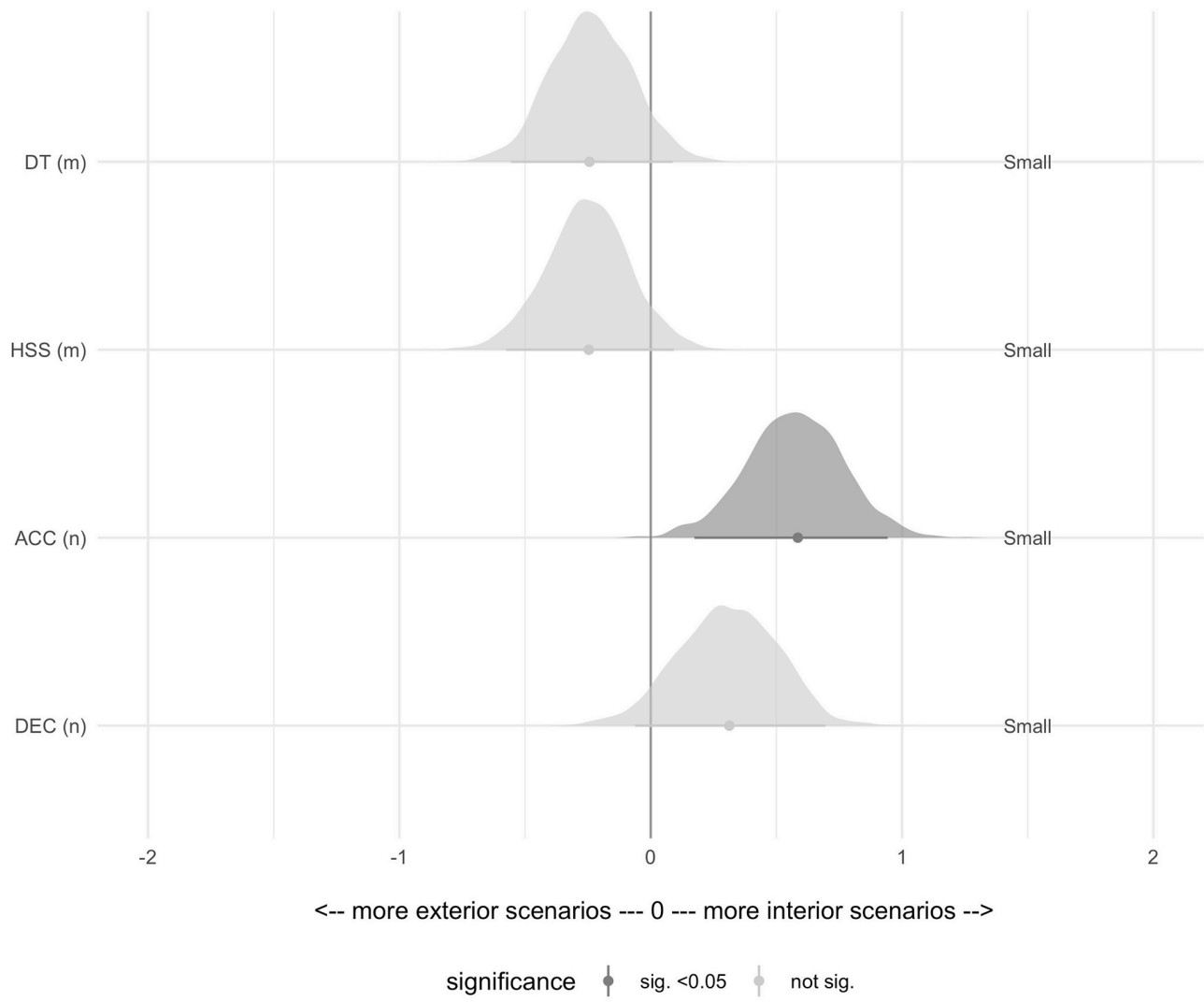

<-- more exterior scenarios --- 0 --- more interior scenarios -->

significance ⬥ sig. <0.05  ⬥ not sig.

**Fig 2. Pairwise Cohen's d differences in the most demanding scenarios (time epoch 30 s) of a match by playing position for each variable.** The significant different variables (high-intensity accelerations) are shown in dark grey, and the non-significant (distance travelled, high speed-skating and high-intensity decelerations) in light grey. Abbreviations—ACC: high-intensity accelerations; DEC: high-intensity decelerations; DT: distance travelled; HSS: high-speed skating; MDS: most demanding scenarios.

of competition, and will help them design more accurate training drills that achieve peak demands and mitigate the risk of injury. Individual training sessions can be an example of how MDS knowledge can be useful. The creation of tasks that simulate different MDS that can be found in matches using different variables can be another way (obviously, not the only one) to schedule and prepare exercises. This enhances the training sessions.

After comparing the peak and average demands of rink hockey, it could be interesting to perform comparisons with other team sports to examine rink hockey in its wider context. The peak demands in rink hockey were higher than in outdoor sports like soccer [11] and rugby [12], except for DT and HSS, where the demands were similar. For indoor sports like basketball [6, 13] and futsal [14] we found peak demands in rink hockey to be greater for all the analysed metrics, except for ACC and DEC in futsal, where the demands were similar. In comparison with outdoor sports, the results of indoor sports are more interesting because of

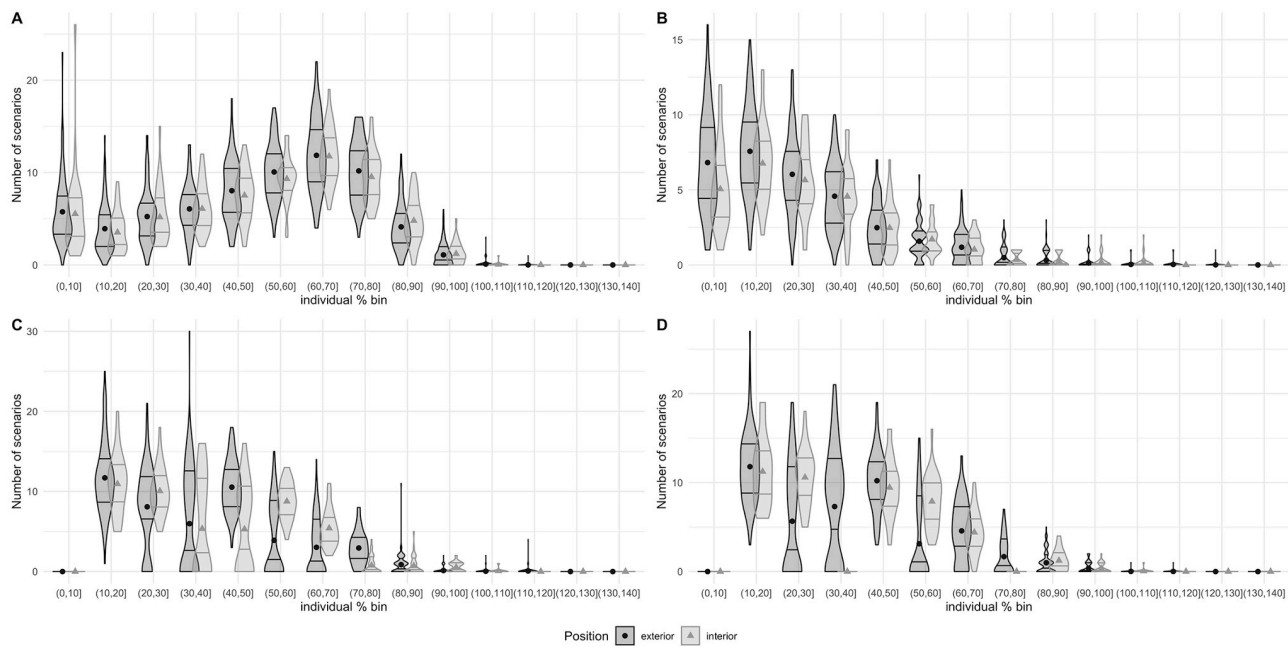

**Fig 3. There is a graphical descriptive number of 30 s scenarios (mean, quartile 1 and quartile 3) in a match for each intensity bucket for each variable.** Exterior players are shown in dark grey and interior players in light grey. A: distance travelled; B: high-speed skating; C: high-intensity accelerations; D: high-intensity decelerations.

the significant differences in the number of players or the space compared with outdoor sports. As reported by Fernández et al. [1], the use of roller skates and fences in rink hockey could account for the higher average values and peak demands of DT and HSS compared to basketball and futsal.

Finally, we found that the MDS data are position-dependent for the ACC metric, albeit with small magnitudes of difference. Interior players performed a greater number of accelerations (from 5.81 to 6.09 ACC per 30s scenario between interior and exterior players) in the MDS of matches. This significant difference may be relevant when prescribing a training programme, despite the low magnitude of effect. As with other indoor sports, the inherent requirements of each position in games and a team's playing model could explain these results [25]; there could be less space for interior players in rink hockey, so they may require more reception and rebounding movements, or press after losing possession of the ball in another court, defend in own court, or support in spaces with high player density by doing more change-of-directions, stops, and other sharp movements. These difference have not been reported for the average demands of competition, where the only published research to date found no significant differences between positions in absolute or relative variables because of the fast and dynamic nature of the sport studied [2]. Therefore, it may be that no significant differences exist in accumulated external load between exteriors and interiors, but that these demands are sensitive to the player's position during the MDS of competition but in a small magnitude. This information can help coaches improve their training tasks. With a range of options as large as adding conditioning blocks focused on the specific demands of specific positions (i.e., interior players), up to changing rules, number of players or spaces in small-sided games (i.e., reducing the space of the court as literature about small-sided games demonstrates that is a variable that highlights the ACC and DEC demands [26]).

**Table 3. Descriptive results of the repetitions number of 30 s scenarios by interior and exterior players for each bucket and variable per match.**

| Position | Bucket | DT | HSS | ACC | DEC |
|---|---|---|---|---|---|
| Interior | (0,10) | 5.53 ± 4.48 | 5.06 ± 2.70 | 0.00 ± 0.00 | 0.00 ± 0.00 |
| | (10,20) | 3.53 ± 2.14 | 6.76 ± 2.40 | 10.9 ± 3.58 | 11.8 ± 4.01 |
| | (20,30) | 5.18 ± 2.94 | 5.65 ± 2.37 | 10.1 ± 2.89 | 5.66 ± 6.02 |
| | (30,40) | 6.09 ± 2.47 | 4.56 ± 2.09 | 5.35 ± 6.11 | 7.30 ± 6.24 |
| | (40,50) | 7.53 ± 2.64 | 2.47 ± 1.58 | 5.29 ± 5.39 | 10.2 ± 3.20 |
| | (50,60) | 9.32 ± 2.38 | 1.71 ± 1.06 | 8.76 ± 2.22 | 3.12 ± 4.57 |
| | (60,70) | 11.8 ± 2.86 | 1.03 ± 0.90 | 5.41 ± 2.23 | 4.57 ± 3.28 |
| | (70,80) | 9.53 ± 2.68 | 0.35 ± 0.49 | 0.76 ± 1.10 | 1.68 ± 2.00 |
| | (80,90) | 4.79 ± 2.46 | 0.26 ± 0.45 | 0.71 ± 1.14 | 0.99 ± 1.19 |
| | (90,100) | 1.21 ± 1.15 | 0.18 ± 0.46 | 0.53 ± 0.61 | 0.38 ± 0.63 |
| | (100,110) | 0.06 ± 0.24 | 0.12 ± 0.41 | 0.09 ± 0.29 | 0.02 ± 0.14 |
| | (110,120) | 0.00 ± 0.00 | 0.00 ± 0.00 | 0.00 ± 0.00 | 0.02 ± 0.14 |
| | (120,130) | 0.00 ± 0.00 | 0.00 ± 0.00 | 0.00 ± 0.00 | 0.00 ± 0.00 |
| Exterior | (0,10) | 5.75 ± 3.45 | 6.82 ± 3.23 | 0.00 ± 0.00 | 0.00 ± 0.00 |
| | (10,20) | 3.92 ± 2.58 | 7.57 ± 2.86 | 11.7 ± 4.38 | 11.2 ± 3.76 |
| | (20,30) | 5.24 ± 2.89 | 6.04 ± 2.53 | 8.09 ± 5.03 | 10.6 ± 3.08 |
| | (30,40) | 6.07 ± 2.64 | 4.58 ± 2.18 | 5.98 ± 6.46 | 0.00 ± 0.00 |
| | (40,50) | 8.04 ± 3.22 | 2.49 ± 1.64 | 10.5 ± 3.15 | 9.44 ± 3.10 |
| | (50,60) | 10.1 ± 2.99 | 1.58 ± 1.19 | 3.90 ± 4.55 | 7.88 ± 2.83 |
| | (60,70) | 11.9 ± 3.86 | 1.18 ± 1.09 | 3.03 ± 3.47 | 4.41 ± 2.31 |
| | (70,80) | 10.2 ± 3.16 | 0.49 ± 0.74 | 2.94 ± 2.02 | 0.00 ± 0.00 |
| | (80,90) | 4.12 ± 2.51 | 0.31 ± 0.56 | 0.87 ± 1.30 | 1.24 ± 1.18 |
| | (90,100) | 1.10 ± 1.24 | 0.13 ± 0.37 | 0.13 ± 0.37 | 0.32 ± 0.53 |
| | (100,110) | 0.10 ± 0.38 | 0.04 ± 0.19 | 0.05 ± 0.25 | 0.06 ± 0.24 |
| | (110,120) | 0.01 ± 0.10 | 0.04 ± 0.19 | 0.08 ± 0.43 | 0.00 ± 0.00 |
| | (120,130) | 0.00 ± 0.00 | 0.01 ± 0.10 | 0.00 ± 0.00 | 0.00 ± 0.00 |

Note. Data shown means ± standard deviations. Abbreviations—ACC: high-intensity accelerations; DEC: high-intensity decelerations; DT: distance travelled; HSS: high-speed skating.

## Quantification of scenarios' repetition relative to individual MDS and the differences between positions

To assess repetition, we divided the scenarios by relative intensity of an individual MDS. The resulting scenarios were categorised as low demand (<40% of individual MDS), moderate demand (40%–80%, MDS), high demand (80%–90%, MDS), and very high demand (>90%, MDS), according to previously published criteria [16]. This allowed us to determine the repetition of 30 s scenarios with different levels, depending on the intensity and the variable under investigation. Similar to our results, research regarding official futsal matches [15] concluded that high-demand and very high-demand scenarios occurred more than once in a variable-dependent manner. Results from basketball matches [16] indicates that scenarios recur across all intensities to different extents. Finally, research regarding rugby league and Australian rules football [17, 18] showed that scenarios of a moderate relative intensity recurred most during official matches.

Our methodology encourages a practical in-field use by counting the number of scenarios in each bucket across the intensity spectrum. This could be useful for providing coaches with reference data on the number of repetitions to programme into training drills to under- or

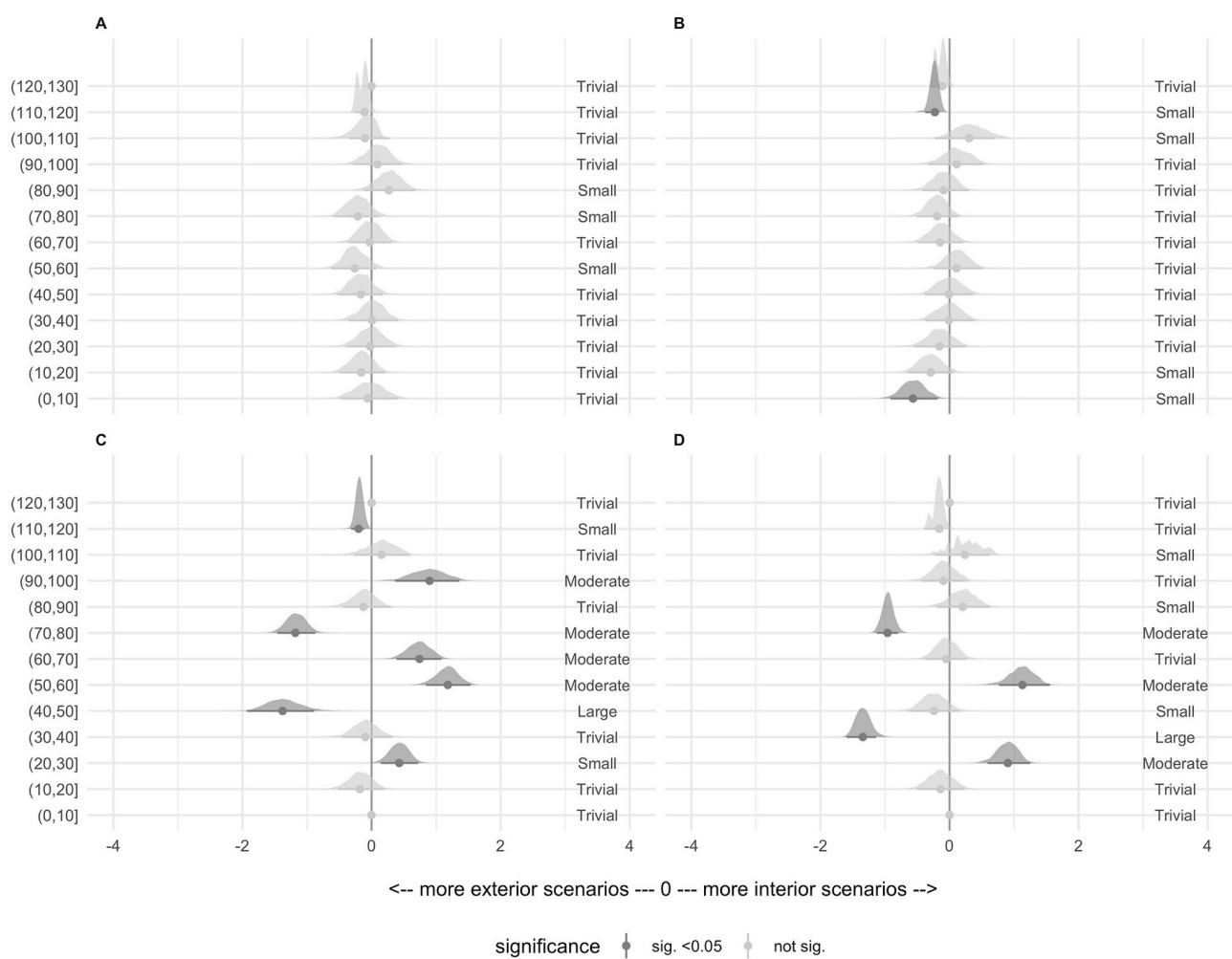

**Fig 4. Pairwise Cohen's d differences between positions for each match bucket per variable per 30 s.** The buckets with significant differences between positions are in dark grey, and the non-significant differences are in light grey. A: distance travelled; B: high-speed skating; C: high-intensity accelerations; D: high-intensity decelerations.

over-stimulate the demands of the competition (e.g., doing 6 reps of a 30 s drill at approximately 70%–80% of an individual MDS DT will under-stimulate the match volume requirements in terms of scenario repetition). Volume-related variables (DT) had the greatest proportion of scenarios between 50% and 90% of the individual MDS, with a peak in the 60%–70% bucket. The higher the intensity criterion of the studied metric, the greater the proportion of scenarios that seem to accumulate in lower buckets. For example, the peak scenario repetition was in the 10%–40% bucket for ACC and DEC compared to the 10%–20% bucket for the HSS. So, in the analysed team, moderate- to high-demand scenarios were repeated most for volume-related variables, while low- to moderate-demand scenarios were repeated most for intensity-related variables.

When comparing the distribution of different scenarios, variables, and playing position, we observed multiple significant differences between exterior and interior players. Like the MDS results, our data revealed that exterior and interior players tend to be exposed to almost the same physical demands in low, moderately high, and very high-intensity buckets for the DT and HSS variables (except in two buckets of HSS). As with the MDS, one explanation could be

that exterior and interior players had a fast, dynamic, and substantial exchange of positions in certain moments of matches. By contrast, interior and exterior players had stochastic differences in the number of scenario repetitions in ACC and DEC metrics only in most of the buckets. Again, as was the case for the MDS, interior players tend to perform more short actions in a confined space, which influences the number of moderate- and high-demand repetitions between interior and exterior players; but this tactical behaviour (short, fast, and sharp actions) does not affect in a clear way the results of our study. It could be that the study of different time epochs is necessary to find some more clear differences between player positions. To the best of our knowledge, this is the first study in rink hockey to consider playing positions when examining how match physical activities are distributed in relation to the MDS. Our results show that the distribution of physical activities seems not to be position-dependent during official competition by professional rink hockey players.

## Limitations

This research has some limitations. First, we collected data from only one professional team, resulting in a small, but exclusive, international, and professional sample. Second, we only examined the MDS and physical activity distribution relative to peak demands for 30 s. As shorter and longer time periods are also possible during a match, using 60, 120, 180, and 300 s epochs might yield more useful data. Third, this is a field pioneering research on the repetition of different intensity scenarios. Consequently, some procedures, such as calculating individual reference values, using intensity buckets, or using fixed time epochs, may require more investigation to find the best method that would allow a more accurate description of the competition. The distribution of preparatory activities regarding the MDS of a match during different training sessions (1, 2, 3, and 4 days prior to competition) should be considered in future research, as well as methodology improvements, to optimize the performance of individual players and teams.

## Conclusions

In conclusion, this research shows that rink hockey matches have higher 30 s peak demands than the averages reported in other research [2], and that these peak demands are position-dependent, with greater ACC values for interior players. Rink hockey matches also appear to include multiple exposures to (very) high-demand scenarios that are close to the peak physical demands of a match. We detected different scenario repetition distributions across all intensity buckets among the variables studied. Therefore, coaches and sport science professionals should consider the repetition of scenarios during training sessions based on the results. On individual interventions for example, interior players may also need a training stimulus increase for the ACC metric. Depending on the goal of training and the context within the season, adding more repetitions or a higher relative intensity based on the individual MDS may be appropriate.

From a practical perspective, the repetition of scenarios application allows coaches to program different intensity sets of fixed time epochs (which must be prescribed using the individual MDS reference) and determine whether the player is performing a volume below, equal, or above match demands. Moreover, this methodology allows to control the external load demands of the weekly training cycle, by using different intensity buckets (e.g., low, moderate, and intense) to determine which training session causes a greater repetition of intense scenarios, and whether it is above, equal, or below match demands. Alongside the lack of previous literature on this topic, our findings should stimulate the external load control future research, specifically, in rink hockey.

## Supporting information

**S1 File.**
(XLSX)

## Acknowledgments

The authors would like to express our deep gratitude to Núria Fernández Raventós and Marina Valls i García for their support in the manuscript reviewing. The authors gratefully acknowledge the help of all elite players who participated in the study. The authors also would like to thank all the members of the sports science area of FC Barcelona for their valuable contribution to this study. The authors finally thank the collaboration of TecnoCampus for their valuable contribution to this study.

## Author Contributions

**Conceptualization:** Daniel Fernández, Joan A. Cadefau, Gerard Carmona.

**Data curation:** Daniel Fernández.

**Formal analysis:** Daniel Fernández.

**Funding acquisition:** Noemí Serra.

**Investigation:** Gerard Carmona.

**Methodology:** Daniel Fernández, Gerard Carmona.

**Project administration:** Gerard Carmona.

**Software:** Daniel Fernández.

**Supervision:** Joan A. Cadefau, Noemí Serra, Gerard Carmona.

**Visualization:** Daniel Fernández.

**Writing – original draft:** Daniel Fernández, Gerard Carmona.

**Writing – review & editing:** Joan A. Cadefau, Noemí Serra, Gerard Carmona.

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
