## [Decision Letter · Decision Letter 0]

6 Dec 2022

PONE-D-22-29744The distribution of different intensity demanding scenarios in elite rink hockey players using an electronic performance tracking system.PLOS ONE

Dear Dr. Carmona,

Thank you for submitting your manuscript to PLOS ONE. After careful consideration, we feel that it has merit but does not fully meet PLOS ONE’s publication criteria as it currently stands. Therefore, we invite you to submit a revised version of the manuscript that addresses the points raised during the review process.

ACADEMIC EDITOR:Dear authors,

Your work is quite interesting, new and deserves to be acknowledge. Nonetheless, there are points to improve before a furthrer recommendation. Please answer both reviewers point by point and highlight your changes in the manuscript.

BR

We look forward to receiving your revised manuscript.

Kind regards,

Rafael Franco Soares Oliveira

Academic Editor

PLOS ONE

Journal Requirements:

Additional Editor Comments:

Dear authors,

Your work is quite interesting, new and deserves to be acknowledge. Nonetheless, there are points to improve before a furthrer recommendation. Please answer both reviewers point by point and highlight your changes in the manuscript.

BR

Reviewers' comments:

Reviewer's Responses to Questions

**Comments to the Author**

1. Is the manuscript technically sound, and do the data support the conclusions?

Reviewer #1: Yes

Reviewer #2: Yes

2. Has the statistical analysis been performed appropriately and rigorously? 

Reviewer #1: Yes

Reviewer #2: Yes

3. Have the authors made all data underlying the findings in their manuscript fully available?

Reviewer #1: Yes

Reviewer #2: Yes

4. Is the manuscript presented in an intelligible fashion and written in standard English?

Reviewer #1: Yes

Reviewer #2: Yes

5. Review Comments to the Author

Reviewer #1: General comments

This article explores the most demanding scenarios according to players’ positioning (internal or external players) and consider it from different intensity scenarios relative to the most demanding individual scenarios.

In general, the study is well written and provide important practical information regarding the physical demands of rink hockey. The study adds important practical knowledge to the field. Therefore, the authors should be congratulated for developing such work.

Despite that, I have some comments that should be addressed.

Specific comments

Abstract

L. 16-24. Please consider using a better connection between what research has explored and study aim. As it is, the idea that brings is that research explored a new approach to analyze physical performance, then the study aim is…

L.32-34. I would suggest having a conclusion more specific. That is, what coaches / sport scientists / strength and conditioning professionals can do with the information from exterior and interior players?

Introduction

In the first paragraph, I would recommend strengthening the importance of physical performance in rink hockey. That is, using this paragraph to characterize the physical demands, including of different playing positions and how understanding the game demands may help coaches to plan and prepare the training microcycle. In the second paragraph, then report to the most demanding scenarios.

L. 40. “Roller skating makes rink hockey a dynamic and fast sport that has relative average demands far greater than other indoor sports”. Please provide reference to support this sentence.

L.76-79. Please consider including study hypothesis.

Methods.

Subjects:

L90-91. Why did the goalkeepers not be included? Add additional rationale.

L.94-95. May the study findings be limited by the low sample size and mainly number of players per playing position (n=7 vs n =2?). It was performed the prior sample analysis to state how many players should the study include (G*Power)?

L.124-126. Include reference to state the corresponding thresholds.

Procedures

L103. What does it mean “same environmental conditions”?

Discussion

L.229-230. What interpretations do the authors make from these findings? Does this mean that this variable does not allow to discriminate winning and losing teams?

L.225 and 227. Can the authors be more specific? I mean, how will this information really help coaches in the practice? What will change? Adding this information will help readers in a better understanding.

L.228 – 234. I would recommend some caution in this information. While I understand the intention of compare with other sports, I would suggest doing it with external vs internal sports separately. First because Soccer and Rubgy had a larger number of players involved in the practice, in which, we know that contribute to different physical demands. Then, by the similarity between “environmental conditions”, basketball and futsal may be more identical, and thus, a higher emphasis should be placed on these two sports.

L.236. Figure 1, is it correct in this sentence? Or should be figure 3?

L240-241. May it be that players have lower distance to the nearest teammate and opponents (i.e., interior players), which as consequence, will require to accelerate or deceleration more often, not only as part of rebounding and receiving, but to press, defend, constrain the opposition, provide support in spaces where a higher presence of players may require more change-of-directions, stoppages, etc?

L244-246. Here I would consider, again, important to translate this to the practice. While coaches may tailor different training tasks according to players positioning, in this case, I would consider relevant to highlight how coaches should increase the acc dec demands for this position (which coaches may then suggest using previous research to adopt specific SSG that seems to encourage these movements).

L. 251-259. A follow up study from this information, would be trying to use X and Y data that WIMU makes available to understand which tactical configurations would contribute to such efforts. I mean, usually, the physical requirements of a match are related with tactical factors, such as ball being lost, exploring fast attacks after recovering the ball, etc…

L.261-265. Well done!

L.283 which tactical behaviour?

L.299. Please correct “method-ology”

Reviewer #2: Abstract

Is not clear “…demanding scenario” introduction that allow to highlight the problem, emphasizing the aims studied. If author describe position of playing as a covariable of analysis (stated in the objectives), is important to understand the playing position stratification, that support the results described under (“position-dependent…interior players…exterior players”). In the methods section authors should describe the major variables of analysis, defining the units. This will support the quantitative results that are required/desirable in the abstract section. The conclusion does not stated the final application of the results, regarding the impact on the training process.

Introduction

1. The problem around the already published data is not so clear: “However, the approach used to assess average physical demand underestimates peak intensities during indoor or outdoor matches…” (line 43-45);

2. Is still unclear in what way this new methods of assessment bring new data, allowing further application. Authors need to systemize what the other instruments give and what the new data collection will allow;

3. Authors described the utility of this data for “…sports medicine practitioners…” (line 58), and I think that this is important for “ sport coaches” or “sport scientists”, independently of the role in the sport team;

4. This section needs revision, namely structuring what we have in literature and what this paper brings of new, before describe the final purpose of the study.

Methods

1. We suggest to adjust the design study, as a observational study, with retrospective data collection (line 82-84);

2. About the sampling/subjects, it is not clear if the data is about repetitive measures along the matches, in the same players along the years of follow-up. If so, in the design is necessary to describe this study as a observational study supported in a cohort, with a time of follow up of….years;

3. The stratification in interior and exteriors needs explanation, namely about the characteristics that fit and define the playing position (line 94-95);

4. Please reword “consent” (line 100);

5. Because physical demands were truly dependent of the opponent, how the authors take into account this on match selection?

6. Please check on the manuscript the writing word “meters” and not “metres” (line 122), check in the all manuscript;

7. The authors need to be more clear about the cutpoints or the thresholds used because of the “skating limitation” and citing studies specific of the context or highlighting this on limitation section/discussion; Futsal and basketball, despite of the indoor profile, is not the same because of the skating specificity. How authors think about this, clarifying the readers.

8. The structure of the figure 1 is not clear in the manuscript. The top 3 description do not have into account the playing position? If no, what authors think about some bias on the results?

9. Consistent statistical analysis section in contrast with some confusion in describing the general procedures. Please revise to make more clear to the readers;

Results

1. Is important to be more clear about the units of the data collection, namely if the variables presented were an average per game (supported in the 18 games collected). We know that not every players have the same opportunities about the time of playing. How authors adjust the results regarding this aspect? This have an important implication in all the dimensions of external load, and the authors describe as a team (table 2).

2. Regarding the variables collected, it seems very underestimated results, regarding the most demanding scenarios. How authors explain this (table 2)?

3. For a common reader regarding high-intensity accelerations and high-intensity decelerations, the units of theses variables were not understandable. You read as counts per something. I suggest number of accelerations/decelerations in the intensity described above (methods section) per game, adjusted for the individual playing time;

4. The authors have so many results to summarise that make the manuscript confused for the reader; I suggest a clear paragraph of the evidence from each table, conferring a clear line of results.

Discussion

1. In the first paragraph of this section authors need to be more state to the point, clarifying the answer to the research topic. Between lines 213-215, please revise the paragraph about what authors real found;

2. I suggest clear messages regarding the differences between positions and the expression of the dependent variables on the game, describing the dimensions of the load per game, in this professional level. For example, distance covered per game, by position, at high intensity? Number of high intensity accelerations and decelerations per game, per position, adjusted by playing time? The fluctuation of the load across the professional game?;

3. Authors describe as “results of table 2…” in discussion section. Please revise to “results of this study” or “in this study we observer…” (line 223); Line 236 do not use the word “figure” in this section;

4. Between lines 225-227, this results were expected to help the coaches. How? Please describe, supported on your results, the impact on training process.

6. PLOS authors have the option to publish the peer review history of their article (what does this mean?). If published, this will include your full peer review and any attached files.

Reviewer #1: No

Reviewer #2: No

---

## [Author Response · Author response to Decision Letter 0]

3 Feb 2023

Reviewer #1: General comments

This article explores the most demanding scenarios according to players’ positioning (internal or external players) and consider it from different intensity scenarios relative to the most demanding individual scenarios. In general, the study is well written and provide important practical information regarding the physical demands of rink hockey. The study adds important practical knowledge to the field. Therefore, the authors should be congratulated for developing such work. Despite that, I have some comments that should be addressed.

We are very grateful for the revision and below we are sending the point-by-point response. To follow the changes, we suggest activating the “all-markup” button from the track changes in the word document to follow the line numbers correctly. 

Reviewer #1: Specific comments

Abstract

L. 16-24. Please consider using a better connection between what research has explored and study aim. As it is, the idea that brings is that research explored a new approach to analyze physical performance, then the study aim is…

According to the reviewer, we added a new connection to the sentence between the research already done and the aims of the paper in the L. 16-22.

L.32-34. I would suggest having a conclusion more specific. That is, what coaches / sport scientists / strength and conditioning professionals can do with the information from exterior and interior players?

Thanks to the reviewer, we added a more specific conclusion to the abstract. Explaining in a little bit more detail how coaches could use the paper results in a practical way. Now in the L. 34-36.

Introduction

In the first paragraph, I would recommend strengthening the importance of physical performance in rink hockey. That is, using this paragraph to characterize the physical demands, including of different playing positions, and how understanding the game demands may help coaches to plan and prepare the training microcycle. In the second paragraph, then report to the most demanding scenarios.

As the reviewer points out, we slash the first paragraph in two to highlight the importance of physical performance in rink hockey. Then, in the second paragraph, we added information about MDS.

L. 40. “Roller skating makes rink hockey a dynamic and fast sport that has relative average demands far greater than other indoor sports”. Please provide reference to support this sentence.

We added the needed reference to support this statement, now in the L. 51.

L.76-79. Please consider including study hypothesis.

Following the reviewer comment, we added the study hypothesis in the L. 101-157.

Methods.

Subjects:

L90-91. Why did the goalkeepers not be included? Add additional rationale.

As the reviewer pointed out, we added a brief explanation about why we didn’t include goalkeepers in the study. In the L. 169-170.

L.94-95. May the study findings be limited by the low sample size and mainly number of players per playing position (n=7 vs n =2?). It was performed the prior sample analysis to state how many players should the study include (G*Power)?

To clarify this observation from the reviewer we added an explanation in the L. 257-259. We didn’t conduct any sample size analysis due to the use of the bootstrap method. This method has a set of assumptions that allow the use of a small sample size among other things.

L.124-126. Include reference to state the corresponding thresholds.

In the L. 219 we added the corresponding references.

Procedures

L103. What does it mean “same environmental conditions”?

To clarify that, we added some examples of what it means “same environmental conditions”. Our intention was to explain that there wasn’t any change of court and that all the games were played in the same facility. In the L. 188.

Discussion

L.229-230. What interpretations do the authors make from these findings? Does this mean that this variable does not allow to discriminate winning and losing teams?

We are thankful about that review, but we didn’t understand what’s the point of the revision. Could be that the lines are wrong? In lines between 229 and 230 (now, L. 357 to 361) there aren’t findings or variables about winning and losing teams.

L.225 and 227. Can the authors be more specific? I mean, how will this information really help coaches in the practice? What will change? Adding this information will help readers in a better understanding.

Now, in the L. 352 we added a brief explanation to try to be mor clear and specific about what and how the results could help coaches in the practice. We added more specific information about training changes that can be done with our results.

L.228 – 234. I would recommend some caution in this information. While I understand the intention of compare with other sports, I would suggest doing it with external vs internal sports separately. First because Soccer and Rubgy had a larger number of players involved in the practice, in which, we know that contribute to different physical demands. Then, by the similarity between “environmental conditions”, basketball and futsal may be more identical, and thus, a higher emphasis should be placed on these two sports.

Agree with the reviewer we tried to do some changes to highlight the comparison with indoor sports. To do that we split the explanation into parts, doing more emphasis in the part of indoor sports. Now, in the L. 357 – 363.

L.236. Figure 1, is it correct in this sentence? Or should be figure 3?

Thanks for the correction, the text should be “Figure 3”. We corrected the sentence but, with the comments of the other reviewer, we had to quit the reference to that figure.

L240-241. May it be that players have lower distance to the nearest teammate and opponents (i.e., interior players), which as consequence, will require to accelerate or deceleration more often, not only as part of rebounding and receiving, but to press, defend, constrain the opposition, provide support in spaces where a higher presence of players may require more change-of-directions, stoppages, etc?

We’re agree with the reviewer and his/her hypothesis about the demands of interior players. To improve our manuscript, we used that example in the text. In the L. 371 – 388, we added information about how can be possible the differences in some variables between positions.

L244-246. Here I would consider, again, important to translate this to the practice. While coaches may tailor different training tasks according to players positioning, in this case, I would consider relevant to highlight how coaches should increase the acc dec demands for this position (which coaches may then suggest using previous research to adopt specific SSG that seems to encourage these movements).

Thankfully, to improve our manuscript, we have added a better explanation about how coaches could increase some demands for some relevant player positions. In the L. 393 – 398, we added a paragraph to clarify that.

L. 251-259. A follow up study from this information, would be trying to use X and Y data that WIMU makes available to understand which tactical configurations would contribute to such efforts. I mean, usually, the physical requirements of a match are related with tactical factors, such as ball being lost, exploring fast attacks after recovering the ball, etc…

We think the same about this comment: tactical factors, and game models are the main contribution to the effort and demands of the players. But we think that this part of the discussion is not suitable for follow-up proposals. In the limitations section there are some proposals about new investigation lines that are interesting.

L.261-265. Well done!

Thank you for that feedback, we tried to transfer that kind of ideas into all the manuscript to upgrade our paper.

L.283 which tactical behaviour?

In the L. 442 we specified the tactical behaviour that it seems that it was not clear.

L.299. Please correct “method-ology”

Now in the L. 462, that word is corrected.

 

Reviewer #2: General comments

Abstract

Is not clear “…demanding scenario” introduction that allow to highlight the problem, emphasizing the aims studied. If author describe position of playing as a covariable of analysis (stated in the objectives), is important to understand the playing position stratification, that support the results described under (“position-dependent…interior players…exterior players”). In the methods section authors should describe the major variables of analysis, defining the units. This will support the quantitative results that are required/desirable in the abstract section. The conclusion does not stated the final application of the results, regarding the impact on the training process.

We are grateful for the comments and revisions from the reviewer to try to upgrade that manuscript. To follow the changes, we suggest activating the “all-markup” button from the track changes in the word document to follow the line numbers correctly. 

We added a few lines to the abstract to optimize the existing content and adapt it to the reviewer's demands and comments. In line 16, we complete the existing information with some lines about what is the most demanding scenario to highlight the problem. Then, in lines 26-27, we added some information about the used positions to understand better the results section. In lines 28-29, we included the units of the variables used. Finally, in lines 34-36, we added information about the final application of the results, regarding the training implication. After these changes, we want to point out that the abstract only could have 300 words, so we had to do some changes in other parts of the structure to merge the reviewer comments and the existing abstract.

Introduction

1. The problem around the already published data is not so clear: “However, the approach used to assess average physical demand underestimates peak intensities during indoor or outdoor matches…” (line 43-45)

We appreciate that revision. To clarify the problem around the already published data and how the average physical demands underestimate peak intensities we added some lines of explanation (line 65-67).

2. Is still unclear in what way this new methods of assessment bring new data, allowing further application. Authors need to systemize what the other instruments give and what the new data collection will allow

In the introduction section, we tried to be more consistent in the order of the explanations. We did a change of order in one paragraph (starting in line 51 now), to be clearer with the explanation about the average values method, the rolling average method to extract the MDS, and finally the repetition of scenarios.

3. Authors described the utility of this data for “…sports medicine practitioners…” (line 58), and I think that this is important for “sport coaches” or “sport scientists”, independently of the role in the sport team

Thanks to the reviewer for that observation. We agree with the revision, and we change “sports medicine practitioners” to “sport scientists” in line 55.

4. This section needs revision, namely structuring what we have in literature and what this paper brings of new, before describe the final purpose of the study

As we said in point 2 of that point-by-point response, we did a change in the order of the paragraphs in the introduction section. We think that now, we have an introduction structured in 4 parts: (1) what is the actual information in average values of the external load in rink hockey, (2) what are the problems of the average values and how does the MDS solve it, (3) which are the problems of the MDS and how the repetition of scenarios solve it, and (4) which are the aims of this paper.

Methods

1. We suggest to adjust the design study, as a observational study, with retrospective data collection (line 82-84)

We added the term “an observational retrospective study” now in the line 160.

2. About the sampling/subjects, it is not clear if the data is about repetitive measures along the matches, in the same players along the years of follow-up. If so, in the design is necessary to describe this study as a observational study supported in a cohort, with a time of follow up of….years

To try to clarify how the data is obtained we added information about the design and the follow up time. In the line 160.

3. The stratification in interior and exteriors needs explanation, namely about the characteristics that fit and define the playing position (line 94-95)

We included information about the playing positions and the characteristics of interior and exterior players to be more specific about the implications of these. Then, this addition of information is useful to understand the results and discussion section. Now, this information is in the lines 176 – 180.

4. Please reword “consent” (line 100)

We reword the word “consent” and now it’s correct. In the line 185.

5. Because physical demands were truly dependent of the opponent, how the authors take into account this on match selection?

Physical demands are indeed truly dependent on the opponent, so, to mitigate this effect, we did a study of 2 years of duration. With this time frame, we have opponents of all kinds of levels (better, equal, or worse than the analysed team). We didn’t add any explanation about that in the text because we think that the description of the number of matches analysed is enough.

6. Please check on the manuscript the writing word “meters” and not “metres” (line 122), check in the all manuscript

We checked all the repetition of “meters” word and now are correct.

7. The authors need to be more clear about the cutpoints or the thresholds used because of the “skating limitation” and citing studies specific of the context or highlighting this on limitation section/discussion; Futsal and basketball, despite of the indoor profile, is not the same because of the skating specificity. How authors think about this, clarifying the readers.

The “skating limitation” it’s true and it’s different from other sports like futsal or basketball. But, we added some references about rink hockey that uses both: variables and thresholds; and we added some lines about how our threshold selection could be useful in future investigations. Now in the lines 209 – 220.

8. The structure of the figure 1 is not clear in the manuscript. The top 3 description do not have into account the playing position? If no, what authors think about some bias on the results?

When we create figure 1, we tried to sum up the procedures used between lines 226 and 240. As we explained in the manuscript and figure 1, the process of the determination of the individual maximum reference value to then calculate the intensity of the other scenarios is individual. So, after that, we calculate all the scenarios of each player, and then we grouped the results by position to do the statistical process. For example, it’s the same as we used 4 or 5 trials to calculate the maximal HR of each athlete, to then process the HR data from some competitions and after that, we compared elite athletes vs. recreational ones.

9. Consistent statistical analysis section in contrast with some confusion in describing the general procedures. Please revise to make more clear to the readers 

We are really grateful about this comment. The utilization of bootstrapping is difficult to implement, and this observation is a great compliment for us.

Results

1. Is important to be more clear about the units of the data collection, namely if the variables presented were an average per game (supported in the 18 games collected). We know that not every players have the same opportunities about the time of playing. How authors adjust the results regarding this aspect? This have an important implication in all the dimensions of external load, and the authors describe as a team (table 2).

That revision is a key point in the data treatment and the results section. The observation that the repetition scenarios are an average per game, and that it’s important to know if all the players had the same opportunities about the playing time; it’s an important observation in this study. To solve that problem, we conducted a previous test to know if players had a significant difference in the minutes played. We did that from the beginning, but we didn’t include that information in the methods. Now we included a clarification statment in lines 178 – 180.

2. Regarding the variables collected, it seems very underestimated results, regarding the most demanding scenarios. How authors explain this (table 2)?

It’s a good question with a clear answer. The results from that table are values in 30 seconds, so it seems underestimated because to compare with results with a value per minute you must multiply the results by 2.

3. For a common reader regarding high-intensity accelerations and high-intensity decelerations, the units of theses variables were not understandable. You read as counts per something. I suggest number of accelerations/decelerations in the intensity described above (methods section) per game, adjusted for the individual playing time

The unit of the variables related to the number of accelerations and decelerations at high-intensity (above 2 m·s2) is count or number. For the MDS we thought that was easy to understand: X number of high-intensity accelerations per 30s scenario (Table 2). Then, for the repetition of scenarios, the units of all the variables are the number of scenarios, for example, X number of scenarios between 50% and 60% of the MDS of interior players. To clarify that and help the readers to understand that they are looking at the repetition of scenarios, we deleted all the units next to the variables in Table 3 and in the captions of figures 3 and 4.

4. The authors have so many results to summarise that make the manuscript confused for the reader; I suggest a clear paragraph of the evidence from each table, conferring a clear line of results.

In the results section, we already have a paragraph for each table or figure. In line 281 about Table 2: “Interiors had greater ACC with significant but small effect sizes”. In lines 301 to 304 about Table 3 and Figure 3: “The DT variable had a greater number of repetitions in the central area of the buckets (between 60%–70% and 70%–80%); the HSS variable had a repetition peak in the first buckets; and the ACC and DEC variables had the greatest number of the scenarios between the 10% and 40% buckets”. And between lines 322 and 325 about Figure 4: “Exterior players did significantly more scenario repetitions in some buckets for the HSS variable, but with a small effect size. By contrast, both interiors and exteriors showed significant moderate differences in repetition scenarios for the ACC and DEC variables in different buckets and different and unclear directions”. So, we thought that with the existent information in the manuscript, readers could understand the main points of all the outputs of our study.

Discussion

1. In the first paragraph of this section authors need to be more state to the point, clarifying the answer to the research topic. Between lines 213-215, please revise the paragraph about what authors real found

We thank the reviewer for the commentary about the first paragraph of the discussion. We tried to do a better paragraph (between lines 325 and 332) and to be more to the point about the aims of the study. More specifically, we changed the aims to match them to the aims of the introduction; and we added what we found in the results to respond to the questions of the research topic.

2. I suggest clear messages regarding the differences between positions and the expression of the dependent variables on the game, describing the dimensions of the load per game, in this professional level. For example, distance covered per game, by position, at high intensity? Number of high intensity accelerations and decelerations per game, per position, adjusted by playing time? The fluctuation of the load across the professional game?

To clarify the messages regarding the differences between positions, we added clear information using the results obtained in the discussion. More specifically, in lines 378-379 we added the difference between exterior and interior players in the ACC variable.

3. Authors describe as “results of table 2…” in discussion section. Please revise to “results of this study” or “in this study we observer…” (line 223); Line 236 do not use the word “figure” in this section.

When we were writing this paper, we used the word figure or table in the discussion to try to contextualize the reader. After the revision, we tried to not to use those references to the results in the discussion section.

4. Between lines 225-227, this results were expected to help the coaches. How? Please describe, supported on your results, the impact on training process.

Thanks for the observation to help coaches and be more practical. We added some lines to try to explain the impact on the training process of our results between lines 352 and 355.

---

## [Decision Letter · Decision Letter 1]

23 Feb 2023

The distribution of different intensity demanding scenarios in elite rink hockey players using an electronic performance tracking system.

PONE-D-22-29744R1

Dear Dr. Daniel Fernández,

We’re pleased to inform you that your manuscript has been judged scientifically suitable for publication and will be formally accepted for publication once it meets all outstanding technical requirements.

Kind regards,

Rafael Franco Soares Oliveira

Academic Editor

PLOS ONE

Additional Editor Comments (optional):

Dear authors,

Congratulations! My recommendation is to accept your work.

Best regards

Reviewers' comments:

Reviewer's Responses to Questions

**Comments to the Author**

1. If the authors have adequately addressed your comments raised in a previous round of review and you feel that this manuscript is now acceptable for publication, you may indicate that here to bypass the “Comments to the Author” section, enter your conflict of interest statement in the “Confidential to Editor” section, and submit your "Accept" recommendation.

Reviewer #1: All comments have been addressed

Reviewer #2: (No Response)

2. Is the manuscript technically sound, and do the data support the conclusions?

Reviewer #1: Yes

Reviewer #2: (No Response)

3. Has the statistical analysis been performed appropriately and rigorously? 

Reviewer #1: Yes

Reviewer #2: (No Response)

4. Have the authors made all data underlying the findings in their manuscript fully available?

Reviewer #1: Yes

Reviewer #2: (No Response)

5. Is the manuscript presented in an intelligible fashion and written in standard English?

Reviewer #1: Yes

Reviewer #2: (No Response)

6. Review Comments to the Author

Reviewer #1: The authors have addressed all my comments. I am satisfied with the review. Congratulations.

P.S. - question regarding winning / losing it was my mistake. Sorry.

Reviewer #2: (No Response)

7. PLOS authors have the option to publish the peer review history of their article (what does this mean?). If published, this will include your full peer review and any attached files.

Reviewer #1: No

Reviewer #2: No

---

## [Editor Report · Acceptance letter]

27 Feb 2023

PONE-D-22-29744R1 

The distribution of different intensity demanding scenarios in elite rink hockey players using an electronic performance tracking system. 

Dear Dr. Carmona:

I'm pleased to inform you that your manuscript has been deemed suitable for publication in PLOS ONE. Congratulations! Your manuscript is now with our production department. 

Kind regards, 

on behalf of

Prof Rafael Franco Soares Oliveira 

Academic Editor

PLOS ONE